# Epidemiological Overview and Traits into Disorders of the Orbital Walls in North-Eastern Romania

**DOI:** 10.3390/medicina61060953

**Published:** 2025-05-22

**Authors:** Ștefan Gherasimescu, Daniela Șulea, Petrica Florin Sava, Alexandra Carp, Lidia Cureniuc, Mihai Liviu Ciofu, Otilia Boișteanu, Marius Gabriel Dabija, Victor Vlad Costan

**Affiliations:** Surgical Department, Faculty of Medicine, University of Medicine and Pharmacy “Grigore T. Popa”, 700115 Iasi, Romania; stefan-gherasimescu@umfiasi.ro (Ș.G.); petrica-florin.sava@umfiasi.ro (P.F.S.); alexandra.i.carp@umfiasi.ro (A.C.); lidia.cureniuc@umfiasi.ro (L.C.); ciofu_mihai@yahoo.com (M.L.C.); otilia.boisteanu@umfiasi.ro (O.B.); marius.dabija@umfiasi.ro (M.G.D.); victor.costan@umfiasi.ro (V.V.C.)

**Keywords:** orbit, orbital fractures, orbital neoplasms, congenital abnormalities, orbital injuries

## Abstract

*Background and Objectives*: This study aimed to assess the frequency and distribution of facial bone injuries in terms of age, sex, residence, location, and etiology in the north-eastern region of Romania. *Materials and Methods*: This retrospective study was conducted within the Oral and Maxillofacial Clinics of “Sf. Spiridon” Hospital, Iași. The study group included 701 subjects (mean age 41.02 ± 18.45; sex: 603 males and 98 females) who were diagnosed with an orbital wall disorder. The epidemiological data on orbital wall fractures—including sociodemographic features, etiology, and location—were statistically analyzed. *Results*: The prevalence of orbital pathology was 1.47% congenital cases, 1.75% tumors, and 96.7% orbital fractures. The distribution of sex, age group, residence, and orbital localization varied significantly among the three diagnostic categories: tumors, congenital conditions, and fractures. Regarding sex, fractures were significantly more frequent in males (87.0%) compared to congenital cases (70.0%) and tumors (41.7%), while tumors showed a female predominance (58.3%) (*p* < 0.001). Congenital conditions predominantly affected patients under 20 years old (90.0%), whereas fractures were more evenly distributed across age groups, with higher frequencies between 31 and 50 years. In contrast, tumors involving the orbital walls were more frequent in older patients, with 33.3% in the 61–70 age group and 25.0% over 70 years. Regarding orbital localization, fractures were more likely to be bilateral (37.3%) or on the right side (44.3%), whereas congenital orbital defects and orbital tumors were predominantly unilateral and left-sided (70.0% and 66.7%, respectively). Bilateral involvement was rare in congenital cases (10.0%) and absent in tumors. *Conclusions*: The data support the finding that men are significantly more prone to trauma and orbital fractures, especially as a result of interpersonal violence and accidents, highlighting the need for preventive measures tailored to sex and socio-professional context.

## 1. Introduction

The orbit is contained within a complex bone tissue architecture with overlying soft tissue involving crucial anatomical structures [1]. Orbital bony wall defects can be of congenital origin or can result from tumors or trauma.

Orbital bony wall defects of congenital origin can result from anophthalmia, microphthalmia, staphylomas, buphthalmia [2], or orbital venous malformation [3].

Tumors of orbital walls diagnosed in the pediatric population can be benign or malignant, as well as congenital or acquired, and they are often different from those found within the adult population. Most of these are benign neoplasms, but their location and proximity to vital organs can have clinically relevant dermatologic and ophthalmologic implications [4]. The most common malignant tumor in childhood is retinoblastoma, while, in adults, optic glioma accounts for two-thirds of all primary optic tumors [5].

Orbital wall defects (fractures) of traumatic origin are the most common maxillofacial trauma and are closely associated with orbital fractures due to the proximity and structural continuity of facial bones and the orbit, leading to the susceptibility of the orbital zone during trauma to the midface area [6].

Maxillofacial injuries affect the facial bones, dentition, and soft tissues, with their increased frequency and severity attributed to the growing dependence on road transportation and rising socio-economic activity [7,8]. Thus, traumatic pathology has become a significant part of the medical intervention in the maxillofacial area [7,8]. Maxillofacial trauma requiring hospital admission is relatively common, with most cases being of mild to moderate severity, frequently accompanied by associated injuries [9,10,11,12,13,14,15]. A significant proportion of maxillofacial trauma cases involve midfacial fractures, whose distribution appears to be influenced by the etiology of the injury, geographic location, local behavioral patterns, and socioeconomic factors [16,17,18]. Orbital fractures represent a distinct medical and social problem within these injuries because of their complicated anatomy, frequency of occurrence, and the potential functional and aesthetic consequences. A complex classification of orbital fractures was presented by Kunz et al. (2014) [19]. Fractures involving the orbital walls may affect various combinations of orbital bones, depending on the direction and intensity of the traumatic force. Clinically, they are categorized into several distinct types. Orbito-zygomatic fractures occur when the malar bone is the primary site of impact. Naso-orbito-ethmoidal fractures are typically the result of trauma directed to the central upper midface. Internal orbital fractures, often referred to as orbital wall fractures (including blow-out and blow-in types), involve the orbital walls without affecting the orbital rim. Lastly, combined orbital fractures involve the entire orbital framework, affecting the orbital walls and margins [19].

As midfacial fractures have become more frequent in the north-eastern part of Romania in recent decades, especially in young adults and males [16], we must expect a similar trend in the prevalence of orbital fractures. A research group reported that the nature of the ocular injuries in midfacial trauma was influenced by the etiology and type of fracture, as well as by the patient’s sex and age [20]. Midfacial complex fractures are frequently associated with orbital fractures due to the anatomical proximity and the direct transmission of traumatic forces to the delicate structures of the orbit [21].

A deep understanding of orbital anatomy, effective clinical evaluation, and awareness of complications is essential in providing appropriate treatment for patients with orbital fractures [22,23]. A research group reported major ocular injuries in 10.5% of patients with maxillofacial trauma, while minor ocular injuries were identified in 36% of maxillofacial trauma; the highest rate of ocular injuries was identified among patients with isolated orbital floor fractures (46.1%) and zygomaticomaxillary complex fractures (28.2%) [21]. Thus, ophthalmologic assessment is crucial in maxillofacial trauma cases to identify ocular injuries, prevent long-term visual complications, and guide appropriate non-invasive or surgical intervention. The diagnostic accuracy of midfacial and orbital fractures has improved due to the development of high-resolution imaging techniques (e.g., CT scanning), 3D surgical planning, and custom orbital implants [24,25]. Also, AI tools proved to have a high level of accuracy in diagnosing orbital fractures and identifying patients requiring surgical intervention; however, their performance in correctly recognizing cases more appropriate for non-surgical management is still under evaluation [26].

Orbital fractures can be associated with complications such as diplopia, enophthalmos, extraocular muscle entrapment, paresthesia of the infraorbital nerve, and deformity or displacement of the ocular globe or eyelids. A research group reported eyelid lacerations, commotio retinae, and ruptured globe in patients with orbital fractures, with a rate of complications of 15.8% (pulmonary complications, diplopia, renal impairment, venous thromboembolism, wound-associated complications). While 25% of patients required surgical treatment, those who underwent surgical repair had significantly higher rates of pulmonary, wound, and ocular motility events [1,27]. Thorough ocular and neurological evaluation can prevent complications such as ocular injuries, orbital deformities, extraocular muscle imbalance, enophthalmos, blepharoptosis, and potential intracranial involvement.

Successful fracture management requires a detailed understanding of the anatomy and pathophysiology, which will ensure the success of orbital fracture management [1,23]. As more than a quarter of maxillofacial trauma patients with orbital fractures have concurrent ocular injuries, all patients with an orbital fracture should undergo a comprehensive ophthalmologic evaluation. A penetrating mechanism of injury, visual acuity deficits, an afferent pupillary defect on physical examination, and radiographic imaging demonstrating fracture depth involvement of the posterior orbit all indicate subjects at high risk for vision loss and should prompt specific concern for ocular injury assessment [28]. While orbital fractures associated with intracranial involvement require a multidisciplinary approach involving both neurosurgery and ophthalmology, a simple fracture of the orbital roof without extension to the inner table of the skull requires only a superior orbitotomy [1,29]. The multidisciplinary team may include ophthalmologists, maxillofacial surgeons, and even neurosurgeons. Regional differences in alcohol use, urban vs. rural residence, and access to early specialized care can influence the timing of diagnosis or lead to suboptimal early management [1,23].

The aim of this study was to better assess the frequency and distribution of facial bone injuries in terms of age, sex, residence, location, and etiology in the north-eastern region of Romania, in order to adapt and optimize prevention, diagnosis, and treatment methods.

## 2. Materials and Methods

This retrospective study was performed within the Oral and Maxillofacial Clinics of Hospital “Sf. Spiridon” Iasi. The study group included 701 subjects (mean age of 41.02 ± 18.45; sex: 603 males; 98 females) who suffered an orbital fracture. The epidemiological data related to orbital fractures (sociodemographic features, etiology, location) were statistically analyzed.

This study was carried out with the approval of the “Sf. Spiridon” Iasi Emergency Clinical Hospital Ethics Committee, as well as that of the “Grigore T. Popa” University of Medicine and Pharmacy Iasi. This study was performed in compliance with the European General Data Protection Regulation (GDPR).

Orbital wall disorders were diagnosed through a combination of clinical evaluation and imaging. Each patient underwent a computed tomography (CT) scan, which is considered the gold standard for diagnosing orbital fractures and other structural pathologies. The scans were independently evaluated by at least two experienced maxillofacial surgeons. In cases where discrepancies arose between evaluators, a consensus was reached through discussion to minimize interobserver variability and improve diagnostic accuracy.

Patients were included if they had a confirmed diagnosis of an orbital wall disorder—whether for surgical reasons, a tumor, or a congenital condition—and if they had complete medical and imaging records. Patients were excluded if their files were incomplete, their diagnosis was uncertain, or if the record was a duplicate from a follow-up visit. In such cases, only the initial presentation was analyzed for statistical purposes. This approach helped avoid duplication and ensured the integrity and standardization of the “database”.

The majority of orbital wall pathology cases were represented by fractures (96.7%), while tumors (1.75%) and congenital conditions (1.47%) had a much lower prevalence. Orbital fractures were significantly more frequent in males, whereas tumors were more common in female patients.

Oro-maxillofacial trauma refers to injuries involving the oral cavity, jaws (maxilla and mandible), facial bones, and associated soft tissues. These injuries may result from road traffic accidents, falls, assaults, sports, or industrial accidents, and they often involve complex functional and aesthetic impairments.

Midfacial trauma involves fractures or soft tissue injuries within the central part of the face, primarily affecting the maxilla, zygoma, nasal bones, orbital floor, and surrounding structures. It typically includes Le Fort fractures (I, II, III), zygomatic complex fractures, and naso-orbito-ethmoidal (NOE) injuries [19].

An orbital fracture is a break in one or more of the bones surrounding the eye, typically involving the thin orbital floor or medial wall. These fractures often occur due to blunt trauma and can be isolated or part of more extensive facial injuries [19].

Diagnostic criteria for orbital fractures include the following [19]:History of blunt trauma to the periorbital region.Periorbital ecchymosis (bruising) and edema.Diplopia (double vision), especially on upward or lateral gaze.Enophthalmos (posterior displacement of the eye).Restricted extraocular muscle movement, often due to entrapment (e.g., of the inferior rectus).Infraorbital nerve hypoesthesia (numbness in the upper cheek or upper lip).Radiographic confirmation (CT scan is the gold standard, showing discontinuity in the orbital walls or herniation of orbital contents).

The prevalence of orbital fracture within total cases of maxillofacial fractures was determined, as well as the features of patients with orbital fractures, as follows:-Sex (males vs. females);-Age groups (0–20 years; 21–30 years; 31–40 years; 41–50 years; 51–60 years; 61–70 years);-Residence (urban vs. rural);-Location (right orbit; left orbit; bilateral);-Etiological factor.

### Statistical Analysis

The data were analyzed statistically using IBM SPSS Statistics 29.0. Qualitative variables were described using frequencies, while quantitative variables were characterized by means and standard deviations, along with maximum, minimum, and median values. The comparative analysis of age in relation to other demographic characteristics was conducted using Student’s *t*-test and ANOVA, while the comparative analysis of clinical and demographic characteristics based on the three diagnostic locations was performed using the Pearson chi-square test. *p*-values less than 0.05 were considered statistically significant, and *p*-values less than 0.01 were considered highly statistically significant.

## 3. Results

The distribution of maxillofacial cases evaluated in the Oral and Maxillofacial Clinics of Hospital “Sf. Spiridon” Iasi (Romania) is displayed in Figure 1a.

A total of 701 cases with orbital wall pathology was diagnosed. In Figure 1b, we noted the decrease in orbital fractures between years 2015–2024 (from 115 cases in 2015 to only 77 cases in 2024).

Table 1 shows features of patients with orbital fractures according to etiology (trauma, congenital, tumors). The prevalence was 1.47% congenital cases, 1.75% tumors, and 96.7% orbital fractures. The distribution of sex, age group, residence, and orbital localization varied significantly between the three diagnostic categories: tumors, congenital conditions, and fractures. Regarding sex, fractures were significantly more frequent in males (87.0%) compared to congenital cases (70.0%) and tumors (41.7%), while tumors showed a female predominance (58.3%). This difference was statistically significant (*p* < 0.001). Age distribution also differed markedly between the groups (*p* < 0.001). Congenital conditions predominantly affected patients under 20 years old (90.0%), whereas fractures were more evenly distributed across age groups, with higher frequencies between 31 and 50 years. In contrast, tumors were more frequent in older patients, with 33.3% in the 61–70 age group and 25.0% over the age of 70. Although a higher proportion of patients across all groups were from rural areas, the difference in distribution by residence was not statistically significant. Patients from rural areas had the highest proportion of orbital tumors (83.3%), followed by fractures (58.6%) and congenital cases (60.0%). Regarding orbital localization, fractures were more likely to be bilateral (37.3%) or on the right side (44.3%), whereas congenital orbital defects and orbital tumors were predominantly unilateral and left-sided (70.0% and 66.7%, respectively). Bilateral involvement was rare in congenital cases (10.0%) and absent in tumors. These differences did not reach statistical significance (*p* = 0.190). Overall, the analysis highlights notable distinctions in demographic and clinical features among the three groups, particularly in terms of sex and age distribution.

Figure 2a,b show the prevalence of orbital fractures in relation to all cases of maxillofacial and midfacial trauma. The rate of orbital fractures was 3.4% in relation to cases of maxillofacial trauma and 54.3% in relation to cases of midfacial trauma.

Related to age, age group distribution was as follows: 0–20 years (100 patients; 14.3%), 21–30 years (127 patients; 18.1%), 31–40 years (136 patients; 19.4%), 41–50 years (131 patients; 18.7%), 51–60 years (83 patients; 11.8%), and 61–70 years (75 patients; 10.7%) (Figure 3a). Patients from the age group of 31–40 years were the most frequently affected by orbital fractures, followed closely by patients from the age group of 41–50 years and 21–30 years

Related to sex, 98 patients were females, representing 14%, and 603 were males, respectively, 86%. The distribution was 6.14 to 1 in favor of the males (Figure 3b).

Related to residence, 40.9% of subjects were from an urban area, while 59.4% were from a rural area (Figure 3c).

Regarding location of orbital fractures, most cases were found in left orbit (45.1%), followed by right orbit (36.9%), and bilateral location was found in 18% of cases (Figure 4).

Regarding the etiology of orbital fractures, the most frequent factor was interpersonal violence, representing 43.5% of cases, followed by traffic accidents (18.3%), accidental fall (17.6%), animal aggression (7.1%), with work (1%) and sport (1%) accidents the last placed etiological factors (Figure 5).

## 4. Discussion

It is crucial to have an understanding of the epidemiological profile of orbital fractures in a given population as this information can lead to better prevention and management. Although data on orbital trauma at the national and international levels have increased, regional data are still lacking. In the north-eastern part of Romania, oromaxillofacial fractures are a significant health issue, with a high incidence rate among the population. This is likely due to a combination of factors, including the prevalence of high-risk activities and certain factors in the region, such as a high rate of alcohol consumption. In this context, in recent years, there has been an increasing interest in understanding the prevalence and patterns of fractures in the oromaxillofacial regions across different populations.

The prevalence rate of 3.4% from the total number of maxillofacial trauma cases in our present study is lower compared to data reported by other research groups. Thus, 51.1% of individuals with maxillofacial trauma were diagnosed with orbital fractures, either as isolated lesions or in association with other facial fractures. Among these, 12.5% of orbital fracture cases (6.4% of the total trauma cases) sustained isolated fractures of the orbital floor [30].

The demographic data from the literature regarding patients with orbital fractures vary significantly, depending on the area where the studies were conducted. The differences related to sex and age groups of patients with orbital fractures could be explained by socio-economic, cultural, and educational factors.

Related to sex distribution, a clear predominance of trauma cases was observed among males, suggesting a significantly higher exposure of men to trauma-related situations. This distribution can be correlated with the most common etiological factors of trauma.

The results indicate a statistically significant difference in sex distribution depending on the type of condition. Fractures were significantly more frequent in males, which may reflect their higher exposure to trauma through intense physical activity, accidents, or risk-taking behaviors. Congenital conditions also showed a male predominance, though less marked, possibly suggesting a genetic or hormonal component that warrants further investigation. In contrast, tumors showed a female predominance, which may indicate sex-related susceptibility factors such as hormonal, genetic influences, or differences in healthcare-seeking behavior. Males are generally at higher risk of maxillofacial trauma (as well as orbital fractures), due to their greater involvement in the active workforce and increased exposure to risk factors such as motor vehicle use, sports activities, social engagement, and substance use, including alcohol. However, in certain regions, the incidence of maxillofacial trauma among females is also significant, likely reflecting shifts in women’s social behavior. Thus, cultural and socioeconomic factors play a crucial role in shaping sex-related prevalence patterns of maxillofacial injuries [8]. According to Singh et al. (2022), orbital fractures are a common pathology in male patients with maxillofacial trauma, accounting for 82.5%, while females represented 17.5% [30]. Our study highlighted the predominance of young male adults (ages between 20 and 30 years) as the most affected category of people with orbital fractures. Young and middle-aged adults are the most frequently exposed to risk factors associated with orbital fractures, such as intense physical activity, occupational hazards, or involvement in accidents and interpersonal violence. The 0–20 years age group accounted for 14.3% of cases, indicating a significant impact among adolescents as well, possibly due to sports injuries or assault. The frequency gradually decreases in older age groups (over 50 years), which may reflect a reduced exposure to traumatic risks in the elderly population. These results may be linked to an active lifestyle and outdoor activities [31]. Individuals in their third decade of life (ages 20–29) represented 34.4% of the patients with maxillofacial trauma [30]. Ko et al. (2013) found a significant percentage of males affected by orbital fractures (68%), most of them between 18 and 44 years of age [27]. The mean age of patients with maxillofacial trauma varies between 29.9 years in Scotland to 43.9 years in Slovenia [32]. Although children can be affected by craniofacial trauma, the absence of sinus pneumatization, bone elasticity, the thickness of the periosteum, and the retruded position of the face in relation to the cranium support a greater degree of protection against maxillofacial trauma [33]. The level of education, the frequency of alcoholic beverage consumption, and involvement of men in raising animals or other activities within the household can influence the epidemiology of maxillofacial fractures [34].

Regarding the etiology of orbital defects, we noted that the most frequent were fractures related to trauma, followed by tumors and congenital pathology. The leading etiological factor identified was interpersonal violence. This type of trauma is typically more prevalent among men and is often associated with physical conflicts or assaults. Traffic accidents were the second most frequent etiological factor, followed by accidental falls. These types of incidents also tend to occur more frequently in males, particularly in the context of physical or work-related activities. Another relevant aspect includes animal aggression and accidental impact, which may reflect exposure to specific occupational or recreational activities. Sports injuries and work-related accidents were rare, possibly due to case selection criteria. As maxillofacial trauma frequently leads to orbital fractures due to the proximity and structural continuity of facial bones and the orbit [6], etiological factors of maxillofacial injuries are also involved in the onset of orbital defects. The etiology of maxillofacial trauma has continuously evolved over the past three decades and continues to do so, varying according to socioeconomic status, cultural characteristics, geographical location, and age group [35]. Maxillofacial trauma has a multifactorial etiology—including road traffic accidents, accidental falls, assaults, industrial mishaps, sports injuries, and firearm injuries—while its severity and pattern depend on the anatomical site affected, the magnitude, and the direction of the traumatic force [8,9]. The orbit is uniquely susceptible to blunt trauma because of its wafer-thin bony walls, proximity to the globe, and surrounding neurovascular structures. In European countries, interpersonal violence is the most incriminated etiology in maxillofacial trauma, responsible of 39% of cases. Other etiological factors of maxillofacial fractures are falls (31%), traffic accidents (11%), sports-related injuries (11%), and work accidents (3%) [32]. These results represent a mean value for the entire European area; other research groups report different data. Some countries reported different results, depending on specific activities typical of these countries. For example, in Austria (highly frequented for winter sports), daily activities (including falls) were the most frequent etiological factors of maxillofacial trauma (38%), while sports accidents (31%) were the second etiological factor. Interpersonal violence accounted for only 12% of cases, similar to traffic accidents (12%). Work accidents accounted for only 5% of work accidents, and other etiological factors accounted for 2% [36]. In Croatia, the main reason for maxillofacial trauma was falls, followed by aggression for men and traffic accidents for women [37]. The same study also describes a difference regarding age, with interpersonal violence more frequent among young people, with falls more common in adults over 50. A study performed in the Netherlands reported traffic accidents as the most frequent etiological factor, followed by interpersonal violence in the case of men and falls for women [18]. In Italy, the first etiological reason was road accidents (57.1%), followed by interpersonal violence (21.7%) and falls (14.2%), while sports accidents (3.3%) were the least common, less encountered factor for maxillofacial trauma [38]. Traffic accidents were the main etiological factor for maxillofacial trauma in India and Saudi Arabia [39,40,41]. The interpretation of these results must consider the influence of age group, sex, and alcohol consumption on the type and severity of maxillofacial trauma as well as orbital fractures. Meanwhile, in elderly people, accidental fall is the most common etiological factor of fracture, and interpersonal violence and traffic accidents are the main factors in patients aged between 15 and 50 years. Maxillofacial trauma in young adults is closely related to physical violence, while, in women, it is most frequently caused by their life partners. The occupants of vehicles involved in traffic accidents are mostly affected by midfacial fractures (a significant percentage of them involving orbital fractures) [42].

The most common etiological factor of orbital fractures is high-impact trauma, with road traffic accidents as the most common etiologic factor in most countries. Other major etiological factors include violence between people, falls, sports injuries, and work accidents. Such fractures are usually seen in the setting of more extensive midfacial trauma, especially in the orbito-zygomatic and naso-orbito-ethmoidal regions, illustrating the susceptibility of the orbital architecture to localized or diffuse forces. Due to disparities in socioeconomic status, behavioral practices, and safety laws by region and population, the etiology of orbital fracture differs substantially in the epidemiological studies [43,44,45,46].

In our study, we noted a progressive decrease in the orbital fractures over time, a result similar to that observed by a previous study performed in the north-eastern area (Romania), where a progressive decrease in the hospital attendance rate for midfacial fractures was observed over the studied period [16]. These data relate to the significant reduction in hospital attendance for facial trauma, when the year 2019 was compared with the same period of the year 2020 [47]. As in the case of midfacial fractures, this can be attributed to greater compliance with public safety rules (tougher sentences for interpersonal violence and traffic accidents), a reduction in alcohol consumption, and better traffic regulations, as well as the implementation of improved diagnostic protocol by CBCT and the decrease in the frequency of conventional radiographic exams [16].

Our study noted a gradual decline in orbital fracture cases from 2015 to 2020, with a high reduction in incidence in 2020, during the COVID-19 pandemic, and a slight gradual decrease from 2021 to 2024. Post-pandemic trauma trends likely reflect several converging factors. First, the COVID-19 pandemic led to widespread social restrictions, reduced interpersonal interaction, and fewer traffic-related and occupational incidents—all major contributors to facial trauma. This alone may explain the sharp downturn in cases around 2020 [48]. However, the decline continues even in the post-pandemic period, suggesting that longer-term societal changes may also be at play. These include the improved enforcement of traffic safety laws, growing public awareness about interpersonal violence, and increasing urban safety measures. Furthermore, advances in imaging and diagnostic technology, along with shifts in hospital referral patterns, may have influenced the way cases are classified or managed at the clinical level [49].

High-resolution CT imaging and 3D surgical planning now enable the precise visualization and customization of implants prior to intervention [50,51]. Moreover, AI-based diagnostic tools have emerged as highly effective in objectively analyzing fracture patterns and predicting the necessity of surgical treatment [52]. Although our retrospective study did not incorporate these technologies, their integration into future epidemiological research and clinical workflows could transform the management of orbital trauma.

The major limiting factor in this study is the retrospective design. It is easy to introduce errors or omissions in patients’ medical records, so the original number of patients with fractures may be imprecise. Additionally, the selection of cases from only a single hospital (“Sf. Spiridon” Hospital, Iaşi, Romania) inevitably introduces sampling bias. Therefore, our results are probably only representative of this area, and the referral of patients for secondary or tertiary treatment may have led to over-representation. Also, the numbers of certain patient subgroups may be too small to produce reliable statistical comparisons with other categories. Another limitation of this research lies in information bias, such as the under-reporting of less severe cases or those without complete records. Furthermore, discrepancies in the diagnostic methods used and diagnostic criteria throughout the study period also contribute to potential inaccuracies—for instance, different techniques were used to measure the direction of the eyeball at various times, while detailed measurements of fracture depth and thickness may not have been recorded consistently by different observers. One of the limitations of this study was that it did not apply a radiological subclassification of orbital fractures based on the type of fracture (e.g., blow-out vs. blow-in) or location within the orbit (e.g., orbital floor, medial wall). Such distinctions are critical in surgical planning and prognosis; yet, because our data were retrospective, we did not have reliable means to stratify them accurately. In the future, prospective studies should aim for standard classification systems to achieve clinical relevance and comparability. Another major limitation of this study is the absence of clinical outcome data. Functional consequences of orbital fractures—such as diplopia, enophthalmos, infraorbital nerve hypoesthesia, ocular motility restriction, or the need for surgical (re)intervention—were not consistently documented in the retrospective patient records. As such, these crucial aspects of morbidity could not be included in the analysis. Although standard univariate tests (χ^2^, *t*-test, ANOVA) were used to analyze relationships between demographic and clinical variables, no multivariable analyses were performed. This is a limitation, as possible confounding factors could not be controlled statistically. Subsequent studies should employ logistic regression models or other multivariable techniques to identify independent predictors of specific orbital pathology outcomes.

In future studies, we propose that the scope be expanded geographically through multicenter studies. Doing so would yield richer results regarding regional variation in orbital injury rates and a more representative understanding thereof. Prospective study designs should also be favored whenever possible, so as to ensure the validity of findings from large-scale observational platforms. Nevertheless, this will be a costly undertaking. Furthermore, behavioral and socioeconomic factors, such as alcohol consumption and occupation, need deeper clarification with structured questionnaire-based surveys. Future research should examine just how cultural norms and social structures are responsible for the higher risk of orbital fracture among men. Longitudinal follow-up studies are also needed to evaluate long-term outcomes in individuals with orbital fractures—specifically, whether their vision is preserved and whether they experience disfiguring complications due to their condition.

From a public health and preventive policy perspective, the top priority indicated by this study’s results is to take targeted measures against interpersonal violence. Young adult males are the group at highest risk and therefore require particular attention. In rural areas, where the incidence of orbital fractures is higher, public awareness and prevention campaigns to promote safer behavior and encourage timely medical intervention are urgently needed. Traffic safety regulations should be strengthened and strictly enforced in high-risk districts, especially those with a significant number of fractures resulting from motor vehicle accidents. This may account for upwards of one eighth or even as much as a quarter of all cases. In addition, preventive education about maxillofacial trauma should become part of occupational safety programs, especially for sectors in those places with a high injury rate, such as construction and agriculture. Organized efforts across these areas would significantly reduce both the number and effects of orbital fractures.

## 5. Conclusions

Most of orbital pathology was represented by fractures (96.7%), while tumors and congenital conditions had a much lower prevalence. Orbital fractures were significantly more frequent in males, whereas tumors were more common in females. Fractures were relatively evenly distributed among patients aged 21–50, with a peak in the 31–40 age group. Tumors were more frequently seen in elderly patients, especially in the 61–70 age group and over 70. Patients from rural areas had the highest proportion of orbital tumors.

In conclusion, the data support the idea that men are significantly more prone to trauma and orbital fractures, especially as a result of violence and accidents, highlighting the need for preventive measures tailored to sex and socio-professional context.

## Figures and Tables

**Figure 1 medicina-61-00953-f001:**
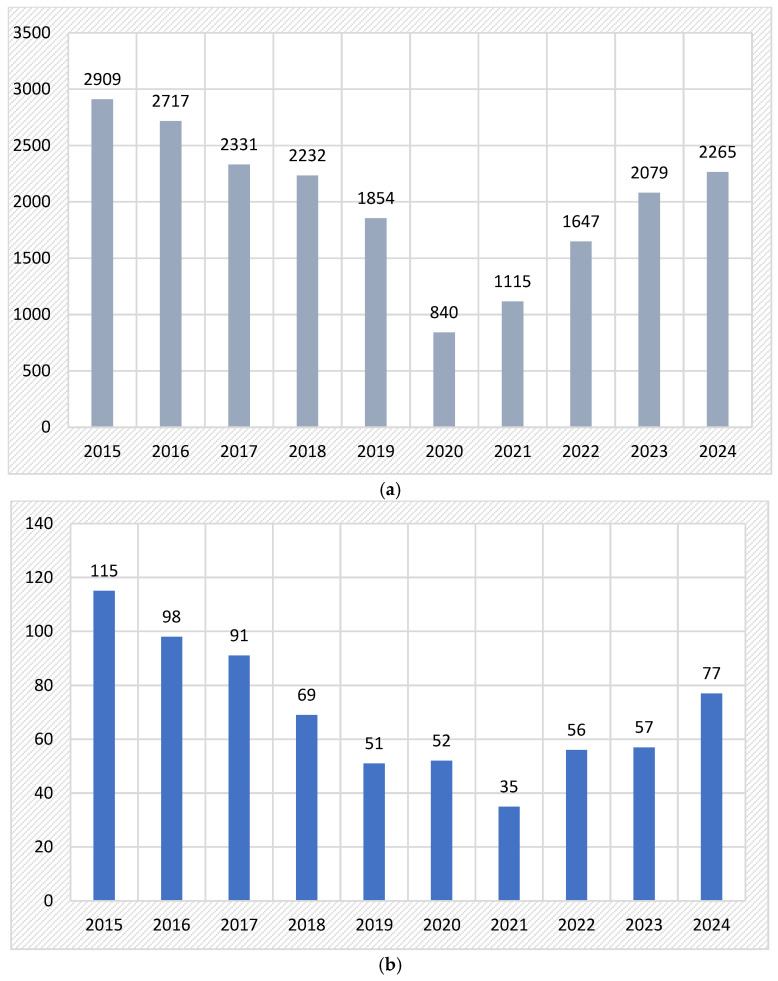
(**a**). The number of cases with maxillofacial pathology over the years of the studied period. (**b**). The number of cases with orbital pathology over the years of the studied period.

**Figure 2 medicina-61-00953-f002:**
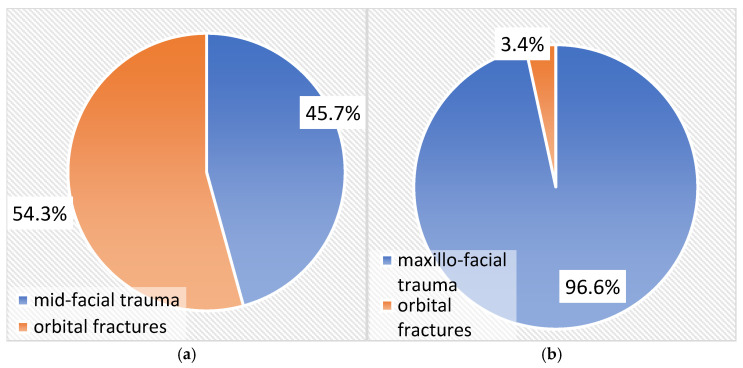
(**a**,**b**). Prevalence of orbital fractures related to maxillofacial trauma (**a**) and midfacial trauma cases (**b**).

**Figure 3 medicina-61-00953-f003:**
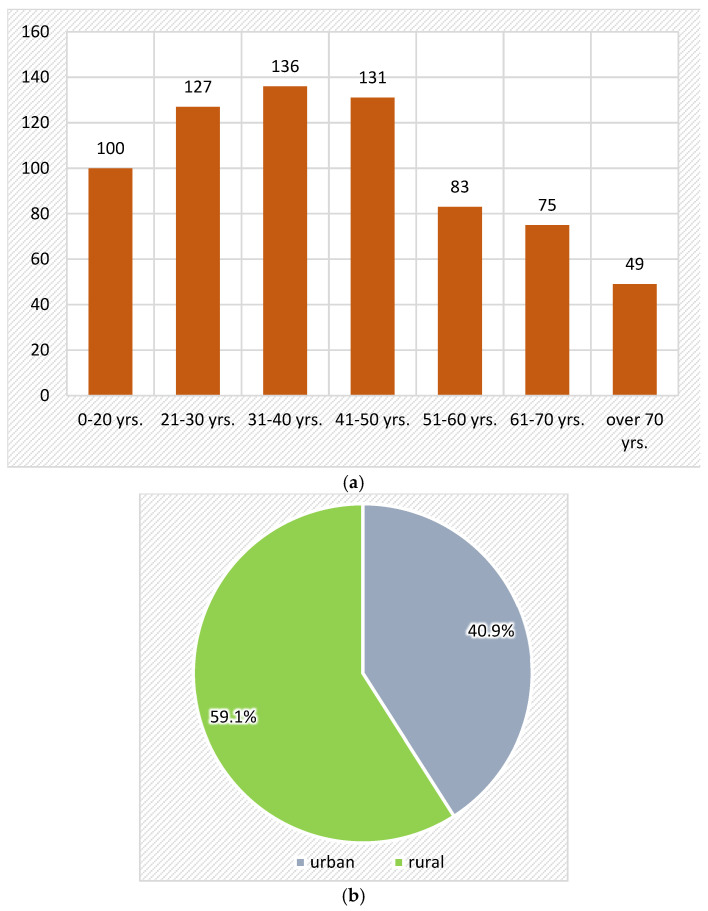
(**a**). Distribution of orbital fractures according to age groups. (**b**,**c**). Distribution of orbital fractures according to sex and residence.

**Figure 4 medicina-61-00953-f004:**
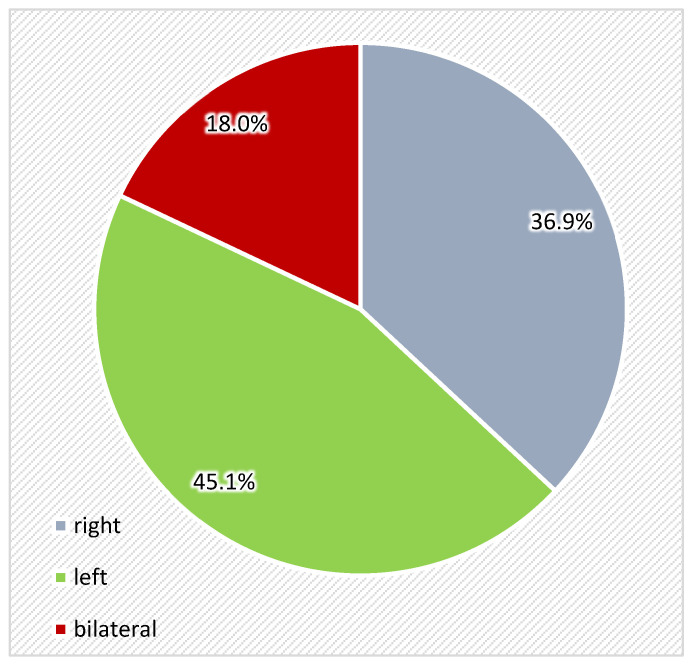
Distribution of the cases with orbital fractures according to location.

**Figure 5 medicina-61-00953-f005:**
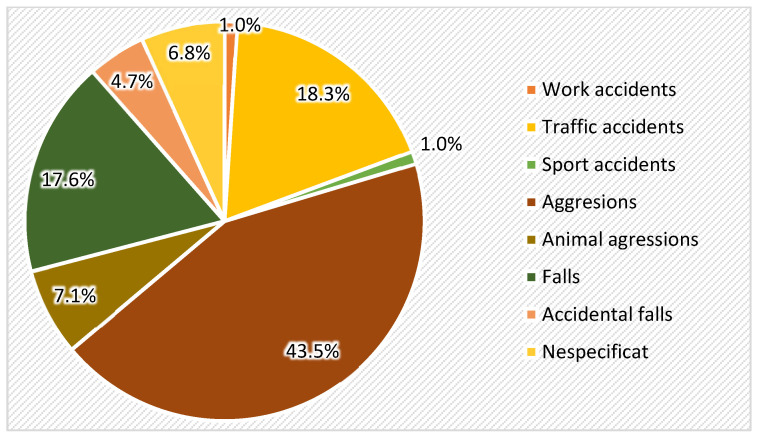
Distribution of the cases with orbital fractures according to etiology.

**Table 1 medicina-61-00953-t001:** Features of patients according to etiological factors of orbital defects.

		Tumors (*n* = 12)		Congenital (*n* = 10)		Fractures (*n* = 679)		Pearson Chi-Squared *p*-Value
		*n*	%	*n*	%	*n*	%	
Sex	Male	5	41.7%	7	70.0%	591	87.0%	<0.001 **
	Female	7	58.3%	3	30.0%	88	13.0%	
Age group	0–20 years	1	8.3%	9	90.0%	90	13.3%	<0.001 **
	21–30 years					127	18.7%	
	31–40 years	1	8.3%			135	19.9%	
	41–50 years	2	16.7%			129	19.0%	
	51–60 years	1	8.3%	1	10.0%	81	11.9%	
	61–70 years	4	33.3%			71	10.5%	
	over 70 years	3	25.0%			46	6.8%	
Residence	Urban	2	16.7%	4	40.0%	281	41.4%	0.225
	Rural	10	83.3%	6	60.0%	398	58.6%	
Orbit location	Bilateral			1	10.0%	253	37.3%	0.190
	Right	4	33.3%	2	20.0%	301	44.3%	
	Left	8	66.7%	7	70.0%	125	18.4%	

** highly statistical significance.

## Data Availability

The data presented in this study are available upon request from the corresponding author. The data are not publicly available due to privacy or ethical restrictions.

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
