# Peer review of "Epidemiological Overview and Traits into Disorders of the Orbital Walls in North-Eastern Romania"

_medicina, 2025, doi:10.3390/medicina61060953_

Round 1
Reviewer 1 Report
Comments and Suggestions for Authors
This study aimed to better assess the frequency and distribution of the facial bone injuries in terms of age, sex, residence, location, etiology in the north-eastern region of Romania, in order to adapt and optimize the prevention, diagnosis and treatment methods. The manuscript is very interesting and I recommend a few comments:
- Abstract: Please review if the extension of this abstract is adequate (300 words)
- Keywords: Please review if they are MeSH terms
- Methods: Please review if the manuscript accomplishes the STROBE checklist for observational studies. It is mandatory that the authors provided a file with the checklist.
The variable is sex not gender, since the research is not asking for social and cultural construction about gender identity. Please review all the document.
- Results: Please review if the tables and figures are adequate according to the instruction for authors.
- Discussion: Please specify clearly the limitation of this study, and recommendation for research and practice (policies and strategies) as based on the main findings
I hope this comments and suggestion be useful.
Regards and best wishes.
Author Response
Comments 1: Abstract: Please review if the extension of this abstract is adequate (300 words)
Response 1: Thank you for bringing this to our attention. Abstract was reduced to under 300 words.
Comments 2: Keywords: Please review if they are MeSH terms
Response 2: Thank you for this observation. We replaced keywords with next MesH terms: orbit, orbital fractures, orbital neoplasms, congenital abnormalities, orbital injuries
Comments 3: Methods: Please review if the manuscript accomplishes the STROBE checklist for observational studies. It is mandatory that the authors provided a file with the checklist.
The variable is sex not gender, since the research is not asking for social and cultural construction about gender identity. Please review all the document.
Response 3: We thank you for this important observation. In response, we have thoroughly reviewed our manuscript and confirmed that it complies with the STROBE (Strengthening the Reporting of Observational Studies in Epidemiology) checklist for observational studies. As requested, we have completed and uploaded the STROBE checklist as a supplementary file. Each item is referenced with the corresponding page or section in our manuscript to ensure transparency and alignment with best reporting practices.
To address all items of STROBE checklist we introduced in Discussions section the next text regarding the study limitations and potential bias as follows: "The major limiting factor in this study is that it is looking at something that already happened. It is easy to introduce errors or omissions in patients’ medical records, so the original number of fracture patients may be imprecise. Additionally, the selection of cases from only a single hospital ("Sf. Spiridon" Hospital, IaÅŸi) leads inevitably to bias, in this case sampling. Therefore, our results are probably only representative of this area, and referral of patients for secondary or tertiary treatment will have resulted in over-representation. Also, the numbers of certain patient subgroups may be too small to produce reliable statistical comparisons with other categories. Another limitation of this research lies in information bias, such as under-reporting of less severe cases or those without complete records. Furthermore, discrepancies in the diagnostic methods em-ployed and diagnostic criteria throughout the study period also contribute to potential inaccuracies—for instance, different techniques were used to measure the direction of the eyeball at various times, while detailed measurements of fracture depth and thickness may not have been recorded consistently by different observers." (page number 11, paragraph 5, lines 446-459)
Thank you for noticing this detail, which contributes to a better expression within our study. Therefore, we replaced all terms "gender" with term "sex".
Comments 4: Results: Please review if the tables and figures are adequate according to the instruction for authors.
Response 4: Thank you for bringing to our attention that the study contained low-quality images. This will lead to a significant improvement of the article. Therefore, we have made the necessary adjustments to ensure the figures are of higher quality.
Comments 5: Discussion: Please specify clearly the limitation of this study, and recommendation for research and practice (policies and strategies) as based on the main findings
Response 5: Thank you for pointing out this very important aspect and for the guidance you are offering us. We introduced the next recommendation for research and practice (policies and strategies) as based on the main findings as follows:
"In future studies, we propose that the scope be expanded geographically through multicenter studies. Doing so would yield richer results regarding regional variation in orbital injury rates and a more representative understanding thereof. Prospective study designs should also be favored whenever possible, so as to ensure the validity of findings from large-scale observation platforms. Nevertheless, this will be a costly undertaking. Furthermore, behavioral and socioeconomic factors, such as alcohol consumption and occupation, are little studied and need deeper clarification with structured questionnaire-based surveys. Especially significant is the attention to gender disparity observed in orbital injuries. Future research should examine just how cultural norms and social structures are responsible for these higher risks to men. Longitudinal follow-up studies are also needed to evaluate outcomes over time for people with fractures in their orbits: whether they can see well, and whether they suffer disfiguring complications because of their condition. From a public health and preventive policy perspective, the top priority indicated by this study's results is to take targeted measures against violence between individuals. Young adult males are the group at highest risk; therefore, they require particular attention. In rural areas where the incidence of orbital fractures is higher, public awareness and prevention campaigns to promote safer behavior and encourage timely medical intervention are urgently needed. Traffic safety regulations should be strengthened and strictly enforced in high-risk districts, especially those with a considerable fraction of fractures resulting from motor vehicle accidents. This may account for upwards of one eighth or even as much as a quarter of all cases. In addition, preventive education about maxillofacial trauma should become part of occupational safety programs, especially for sectors in those places with a high injury rate, like construction and agriculture. Organized efforts spanning across these areas would make a serious dent in both the numbers and effects of orbital fractures.” (page number 12, paragraph 2, lines 475-499)
Reviewer 2 Report
Comments and Suggestions for Authors
The article “Epidemiological Overview and Traits into Disorders of the Orbital Walls in North-Eastern Romania”addresses a relevant topic by analyzing orbital wall pathologies over a ten-year period. While the dataset is extensive and the focus on a geographically underrepresented region is commendable, the manuscript suffers from a few methodological and conceptual weaknesses that limit its scientific impact.
1. The diagnostic categorization into three broad groups—fractures, tumors, and congenital anomalies—is overly simplistic and lacks clinical granularity. In particular, orbital fractures are not subclassified according to radiological type or anatomical location (e.g., floor vs. medial wall, blow-out vs. blow-in), which are essential for both diagnosis and treatment planning.
2. Another major shortcoming is the absence of clinical outcome data. The study does not report on any functional consequences of the orbital trauma, such as diplopia, enophthalmos, sensory disturbances, or the need for surgical (re-)intervention.
3. The methodology section lacks sufficient detail to allow reproducibility. It is unclear how diagnoses were established—whether by clinical criteria, imaging, or both—and whether all patients received uniform diagnostic imaging (e.g., CT scans). There is also no mention of exclusion criteria, interobserver reliability, or how repeat visits were handled in the dataset.
4. While the authors use standard statistical tests, the analysis remains superficial. There is no multivariate analysis to identify independent predictors for specific outcomes or subgroups. Some results that did not reach statistical significance are still discussed as if they had clinical implications, such as rural vs. urban distribution.
5. The discussion section attempts to provide context but lacks critical engagement with recent international studies. Much of the cited literature is either outdated or descriptive. Contemporary issues such as post-pandemic trauma trends, socioeconomic influences on healthcare access, and advances in treatment (e.g., 3D planning, AI diagnostics) are only briefly mentioned or entirely omitted.
6. From a structural and linguistic standpoint, the manuscript requires substantial revision. The writing contains grammatical errors, repetition, and awkward phrasing. Figures are poorly labeled and lack statistical details (such as p-values or confidence intervals). Additionally, the abstract contains numerical inconsistencies, such as the mention of “1,475% congenital cases,” which appears to be a typographical error. Some of the tables are provided in Romanian.
In conclusion, while the study presents potentially valuable data from an underrepresented population, it currently falls short of the standards required for publication in a high-quality medical journal. The manuscript would benefit greatly from a more detailed diagnostic classification, inclusion of clinical outcomes, clearer methodology, deeper statistical analysis, and a revised discussion informed by current literature.
Comments on the Quality of English LanguageA thorough linguistic and structural revision is also needed.
Author Response
Comments 1: The diagnostic categorization into three broad groups—fractures, tumors, and congenital anomalies—is overly simplistic and lacks clinical granularity. In particular, orbital fractures are not subclassified according to radiological type or anatomical location (e.g., floor vs. medial wall, blow-out vs. blow-in), which are essential for both diagnosis and treatment planning.
Response 1: We thank you for offering this important point of view, which improves our study. Our primary aim was to conduct an epidemiological overview based on available retrospective data, focusing on the frequency and distribution of orbital wall disorders in our region. Given the structure of the hospital records and the limited detail provided in many older entries, a more granular classification (e.g., blow-out vs. blow-in fractures, orbital floor vs. medial wall involvement) was not consistently available across all cases and thus could not be reliably analyzed.
We inserted this limitation of study in Discussions section: "One of the limitations of this study was that it did not apply a radiological subclassification of orbital fractures based on the type of fracture (e.g., blow-out vs. blow-in) or location within the orbit (e.g., orbital floor, medial wall). Such distinctions are critical in surgical planning and prognosis; yet because our data were retrospective, we did not have reliable means to stratify it accurately. In the future, prospective studies should aim for standard classification systems to achieve clinical relevance and comparability." (page number 11, paragraph 5, lines 459-465)
Comments 2: Another major shortcoming is the absence of clinical outcome data. The study does not report on any functional consequences of the orbital trauma, such as diplopia, enophthalmos, sensory disturbances, or the need for surgical (re-)intervention.
Response 2: We thank you for highlighting this important limitation. We fully agree that clinical outcomes—such as diplopia, enophthalmos, sensory deficits, and surgical interventions—are essential to understanding the real-world impact of orbital trauma. However, as this was a retrospective epidemiological study, the primary data available in patient records were limited to demographic characteristics, etiology, and basic localization of fractures. Unfortunately, detailed and consistently recorded information about clinical outcomes or functional impairments was not systematically available for a majority of the included cases. For this reason, outcome data could not be reliably analyzed or reported.
We inserted this limitation of study in Discussions section: "Another major limitation of this study is the absence of clinical outcome data. Functional consequences of orbital fractures—such as diplopia, enophthalmos, infraorbital nerve hypoesthesia, ocular motility restriction, or need for surgical (re)intervention—were not consistently documented in the retrospective patient records. As such, these crucial aspects of morbidity could not be included in the analysis." (page number 11-12, paragraph 5, lines 465-469)
Comments 3: The methodology section lacks sufficient detail to allow reproducibility. It is unclear how diagnoses were established—whether by clinical criteria, imaging, or both—and whether all patients received uniform diagnostic imaging (e.g., CT scans). There is also no mention of exclusion criteria, interobserver reliability, or how repeat visits were handled in the dataset.
Response 3: Thank you for bringing this important aspect to our attention. To address all demands we added next text in Materials and Method section:
"Orbital wall disorders were diagnosed through a combination of clinical evaluation and imaging. Each patient underwent a computed tomography (CT) scan, which is considered the gold standard for diagnosing orbital fractures and other structural pathologies. The scans were independently evaluated by at least two experienced maxillofacial surgeons. In cases where discrepancies arose between evaluators, a consensus was reached through discussion to minimize interobserver variability and improve diagnostic accuracy.
Patients were included if they had a confirmed diagnosis of an orbital wall disorder—whether for surgical reasons, a tumor, or a congenital condition—and if they had complete medical and imaging records. Patients were excluded if their files were incomplete, their diagnosis was uncertain, or if the record was a duplicate from a follow-up visit. In such cases, only the initial presentation was analyzed for statistical purposes. This approach helped avoid duplication and ensured the integrity and standardization of the database." (page number 3-4, paragraph 7, lines 135-147)
Comments 4: While the authors use standard statistical tests, the analysis remains superficial. There is no multivariate analysis to identify independent predictors for specific outcomes or subgroups. Some results that did not reach statistical significance are still discussed as if they had clinical implications, such as rural vs. urban distribution.
Response 4: We thank you for your critical evaluation of the statistical analysis. We acknowledge that our current approach, while based on standard descriptive and comparative tests, remains limited in scope. As suggested, a multivariate analysis (e.g., logistic regression or generalized linear models) would be more appropriate to determine independent predictors of specific outcomes such as fracture type, laterality, or etiology. However, given the retrospective nature of the study and the limitations in variable granularity, such analyses could not be robustly conducted with our current dataset. We have now included this point as a limitation in Discussions section and removed speculative interpretations based on non-significant results, such as urban vs. rural distribution.
To address all demands we added next text in Discussions section (as study limitation):
"Although standard univariate tests (χ², t-test, ANOVA) were conducted to analyze rela-tionships between demographic and clinical variables, no multivariable analyses were done. This is a limitation, as possible confounding factors could not be controlled statis-tically. Subsequent studies should employ logistic regression models or other multi-variable techniques to identify independent predictors of specific orbital pathology outcomes." (page number 12, paragraph 1, lines 470-474)
To address the next concern of this issue we replaced "Concerning the environment, most patients across all categories came from rural areas." with "Although a higher proportion of patients across all groups were from rural areas, the difference in distribution by residence was not statistically significant" (page number 6, paragraph 1, lines 227-229)
Comments 5: The discussion section attempts to provide context but lacks critical engagement with recent international studies. Much of the cited literature is either outdated or descriptive. Contemporary issues such as post-pandemic trauma trends, socioeconomic influences on healthcare access, and advances in treatment (e.g., 3D planning, AI diagnostics) are only briefly mentioned or entirely omitted.
Response 5: We thank you for this important observation. To address this demand we added next text in Discussions section:
“Our study noted a gradual decline in orbital fracture cases from 2015 to 2020, with high reduction of incidence in 2020, in COVID-19 pandemic, and slight gradually decrease from 2021 to 2024. Post-pandemic trauma trends reflects likely reflects several converging factors. First, the COVID-19 pandemic led to widespread social restrictions, reduced interpersonal interaction, and fewer traffic-related and occupational incidents—all major contributors to facial trauma This alone may explain the sharp downturn in cases around 2020 [48]. However, the decline continues even in the post-pandemic period, suggesting that longer-term societal changes may also be at play. These include improved enforcement of traffic safety laws, growing public awareness about interpersonal violence, and increasing urban safety measures. Furthermore, advances in imaging and diagnostic technology, along with shifts in hospital referral patterns, may have influenced the way cases are classified or managed at the clinical level [49].
High-resolution CT imaging and 3D surgical planning now enable precise visualization and customization of implants prior to intervention [50, 51]. Moreover, AI-based diagnostic tools have emerged as highly effective in objectively analyzing fracture patterns and predicting the necessity of surgical treatment [52]. Although our retrospective study did not incorporate these technologies, their integration into future epidemiological re-search and clinical workflows could transform the management of orbital trauma.” (page number 11, paragraph 3, lines 428-445)
Comments 6: From a structural and linguistic standpoint, the manuscript requires substantial revision. The writing contains grammatical errors, repetition, and awkward phrasing. Figures are poorly labeled and lack statistical details (such as p-values or confidence intervals). Additionally, the abstract contains numerical inconsistencies, such as the mention of “1,475% congenital cases,” which appears to be a typographical error. Some of the tables are provided in Romanian.
Response 6: Thank you for bringing this to our attention. Therefore, in order to improve these aspects, we checked and corrected grammar and percentages as well as we labeled properly figures and tables.
Round 2
Reviewer 2 Report
Comments and Suggestions for Authors
The revised manuscript presents an extended and comprehensive epidemiological analysis of orbital wall pathologies in north-eastern Romania, with a clear focus on trauma-related fractures. The authors have made significant efforts to improve the structure, clarity, and interpretative depth of the manuscript, especially in the discussion and limitations sections. The revision has resulted in a more robust and informative article that now offers greater clinical and public health relevance.
Only a few minor points should be addressed:
1. Although improved, there are still numerous grammatical issues and typographical inconsistencies throughout the manuscript (e.g., “gendersex”, “fracturefractures”, “tumor cases was” etc.). A professional proofreading or editorial polishing is strongly recommended prior to publication.
2. Use consistent terminology throughout the text, particularly for epidemiological variables (e.g., “residence” vs. “environment”, “gender” vs. “sex”, “etiology” vs. “cause”).
3. Some statements (e.g., relating to cultural explanations for gender differences) would benefit from more cautious wording or referencing. Suggest phrasing such as “may reflect…” or “could be influenced by…” to avoid overgeneralization.
4. Some paragraphs still repeat findings already shown in results sections (e.g., exact percentages). Consider trimming these to enhance focus on interpretation rather than reiteration.
See above
Author Response
Comments 1: Although improved, there are still numerous grammatical issues and typographical inconsistencies throughout the manuscript (e.g., “gendersex”, “fracturefractures”, “tumor cases was” etc.). A professional proofreading or editorial polishing is strongly recommended prior to publication.
Response 1: Thank you for your helpful observations. Following a detailed grammatical check-up, we eliminated grammatical issues and typographical inconsistencies. We have made every effort to address the details with the utmost care, and we sincerely hope that the revised manuscript now meets the standards you expect. We thank you for taking the time to provide a thoughtful evaluation.
Comments 2: Use consistent terminology throughout the text, particularly for epidemiological variables (e.g., “residence” vs. “environment”, “gender” vs. “sex”, “etiology” vs. “cause”).
Response 2: Thank you for bringing this to our attention. Terminology was checked to ensure consistency.
Comments 3: Some statements (e.g., relating to cultural explanations for gender differences) would benefit from more cautious wording or referencing. Suggest phrasing such as “may reflect…” or “could be influenced by…” to avoid overgeneralization.
Response 3: We are grateful for your insightful remarks. We eliminated phrases related to cultural explanation for gender differences.
Comments 4: Some paragraphs still repeat findings already shown in results sections (e.g., exact percentages). Consider trimming these to enhance focus on interpretation rather than reiteration.
Response 4: We sincerely thank you for your detailed review. We removed from Discussions section all percentages present in Results section.